# Effects of Mixing Conditions on Floc Properties in Magnesium Hydroxide Continuous Coagulation Process

**Yanmei Ding, Jianhai Zhao \*, Lei Wei, Wenpu Li and Yongzhi Chi**

Tianjin Key Laboratory of Aquatic Science and Technology, School of Environmental and Municipal Engineering, Tianjin Chengjian University, Tianjin 300384, China; huaerqi@126.com (Y.D.); weilei900210@163.com (L.W.); hitlwp@126.com (W.L.); 23733403@126.com (Y.C.)

**\*** Correspondence: jhzhao@tcu.edu.cn

**Abstract:** Magnesium hydroxide continuous coagulation process was used for treating simulated reactive orange wastewater in this study. Effects of mixing conditions and retention time on the coagulation performance and floc properties of magnesium hydroxide were based on the floc size distribution (FSD), zeta potential, and floc morphology analysis. Floc formation and growth in different reactors were also discussed. The results showed that increasing rapid mixing speed led to a decrease in the final floc size. The floc formation process was mainly carried out in a rapid mixer; a rapid mixing speed of 300 rpm was chosen according to zeta potential and removal efficiency. Reducing retention time caused a relatively small floc size in all reactors. When influent flow was 30 L/h (retention time of 2 min in rapid mixer), the average floc size reached 8.06 μm in a rapid mixer; through breakage and re-growth, the floc size remained stable in the flocculation basin. After growth, the final floc size reached 11.21 μm in a sedimentation tank. The removal efficiency of reactive orange is 89% in the magnesium hydroxide coagulation process.

**Keywords:** magnesium hydroxide; reactive orange; mixing; coagulation; floc size

---

## 1. Introduction

Magnesium hydroxide was used as a potential coagulant for reactive dyes wastewater treatment for many years [1–3]. Typical characteristics of this kind of coagulant include nontoxicity, an environmentally-friendly nature, recoverability, and rapid reaction [4]. Magnesium hydroxide precipitate has a positive superficial charge and it can adsorb negative dyes through charge neutralization or precipitate enmeshment [5,6]. Floc size settling properties are the main parameters influencing reactive dyes removal efficiency in real industrial scale unit operations [7–9]. A photometric dispersion technique and laser technique are useful in monitoring floc physical characteristics [10,11]. As previously found [12–14], magnesium hydroxide nucleation and precipitation processes are very fast; floc size can reach to 15 μm. Although floc size is not very large compared with other coagulants, the sedimentation process can meet the requirement of pollutant removal.

During the magnesium hydroxide coagulation process, mixing conditions influenced the removal efficiency and floc properties. Rapid mixing brings the reactants together and homogenizes the solution. In the rapid stirring process, it will cause nucleation and precipitation of magnesium hydroxide. The magnesium hydroxide coagulation process had two stages including fast floc formation and growth of flocs. As for slow mixing, flocs remained stable and the re-growth process was not happened apparently in our previously found [13]. Several authors have found that operational conditions influenced the coagulation process and flocs properties especially for mixing conditions [11,15]. Effects of shearing on floc formation and growth are also related to coagulation mechanisms [16,17].

Although there was a large amount of data on the characteristics of floc in the batch coagulation process, there have been limited studies on the relationship between floc characteristics and mixing conditions using magnesium hydroxide as a coagulant in the continuous process. The main objectives of this laboratory study were to evaluate the role of rapid mixing speed on coagulation performance and floc properties. Furthermore, the effects of retention time on floc characteristics are also assessed.

## 2. Materials and Methods

### 2.1. Synthetic Test Water and Coagulant

Reactive orange (K-GN) was purchased from Jinan Xinxing Textile Dyeing Mill, Shandong, China. Reactive dye solutions with pH 12 were prepared with K-GN and deionized water to provide a concentration of 0.25 g/L. NaOH solution was used to control the solution pH value (PHS-25 Shanghai Jinke Industrial Co., Shanghai, China). $MgCl_2 \cdot 6H_2O$ (CP. Tianjin Chemical Reagent Co. Tianjin, China) was used to prepare the coagulant. Magnesium ion was analyzed with an ICS-1500 (Dionex, Sunnyvale, California, USA) ion chromatography system. The concentration of reactive orange in the solution was analyzed by a UV-VISIBLE spectrophotometer (UV2550 Shimadzu, Suzhou, China). The reactive orange characteristics are shown in Table 1.

**Table 1.** Reactive orange characteristics.

| Name | Molecular Structure | $\lambda_{max}$ **(nm)** |
|---|---|---|
| Reactive orange (K-GN) | | 476 |

### 2.2. Apparatus and Procedures

Continuous experiments were carried out at temperature of $20 \pm 1$ °C, and in order to obtain a real continuous steady experiment, K-GN removal efficiency remained stable at 2 h. For this process to occur, the 1 L rapid mixer had to have a stirring speed at 250 to 350 rpm. The flocculation basin was divided into three parts (each of 4 L) and a sedimentation tank (30 L) was designed for the removal of flocs (Figure 1). The slow stirring speed was maintained at 80 rpm in the flocculation basin. 0.25 g/L Reactive orange and 250 mg/L magnesium ion were pumped into the rapid mixer, respectively. The total influent flow was chosen at 30 L/h and 60 L/h, in which retention time was 2 min and 1 min in the rapid mixer, respectively. The operational conditions of the continuous coagulation process are shown in Table 2.

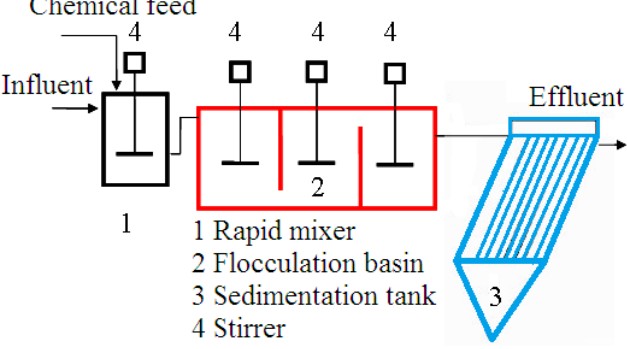

**Figure 1.** Experimental apparatus for coagulation of magnesium hydroxide.

**Table 2.** Operational conditions of coagulation process.

| Flux | Rapid Mixer | | Flocculation Basin | | Sedimentation Tank |
|---|---|---|---|---|---|
| | speed | time | speed | time | time |
| 30 L/h | 250 rpm | | | | |
| | 300 rpm | 2 min | 80 rpm | 24 min | 60 min |
| | 350 rpm | | | | |
| 60 L/h | 300 rpm | 1 min | 80 rpm | 12 min | 30 min |

### 2.3. Floc Size Distribution and Properties Analysis

During the continuous coagulation process, samples of flocs were taken from a rapid mixer, with the third flocculation basin and sedimentation tank using a tube with an inner diameter of 5 mm every 10 minutes. Floc size distribution (FSD) was measured by Mastersizer 2000 (Malvern Panalytical, Malvern, UK) and each sample was measured three times to obtain the average results. During the slow mixing period in the third flocculation basin, zeta potential was measured by zetasizer Nano ZS (Malvern Panalytical, Malvern, UK). The images of flocs from the rapid mixer, third flocculation basin, and sedimentation tank were captured by IX71 digital photomicrography (Olympus, Tokyo, Japan).

## 3. Results and Discussion

### 3.1. Coagulation Behaviors under Different Rapid Mixing Speed

#### 3.1.1. Floc Size Distribution in Three Processes

Continuous experiments were performed under 250 mg/L magnesium ion with 30 L/h to investigate the effects of rapid mixing conditions on coagulation performance and floc size distribution. According to FSD, average floc size decreased when rapid mixing speed increased in the rapid mixer and sedimentation tank. In the flocculation basin, floc size tended to be stable or slightly broken with the increase of rapid mixing speed to 300rpm. This is consistent with the findings that repulsive forces tend to stabilize the suspension and prevent particle agglomeration [13,18]. As shown in Table 3, the average floc sizes 8.39, 8.06 and 8.04 μm were obtained with rapid mixing speeds of 250, 300 and 350 rpm in a rapid mixer, respectively. Small floc had the same trend to aggregate relatively large flocs. Average floc size was 8.06 and 7.89 μm in the rapid mixer and flocculation basin when the mixing speed was 300 rpm. In general, the flocs will grow in the flocculation basin, but it seems that floc size decreased in the flocculation process. In fact, during the flocculation process, particles larger than 11.25 μm accounted for 4.54% and 4.9% in the rapid mixer and flocculation basin, respectively. Particles smaller than 1 μm also decreased in the slow mixing period. As can be seen also in Figure 2 (sedimentation tank), particles smaller than 1 μm account for 6.2%, 10.0% and 12.6% with mixing speeds of 250, 300 and 350 rpm respectively. It was observed that the percentage of smaller particles increased with the increase of mixing speed. This indicates that the high mixing speed will break the flocs, and only part of the flocs will aggregate again after the flocs are broken. The general shape of the curves was broadly similar in different units. The average floc size decreased to a steady state; there was a dynamic balance between floc growth and breakage. When the stirring speed was increased from 250 rpm to 350 rpm, small average floc size could be observed in these three units. Following the breakage and aggregation period, there was only a partial flocs re-growth, showing that the broken flocs can only re-grow to a very limited extent.

**Table 3.** Average floc size in different process units.

| Rapid Mixing (rpm) | Average Floc Size (μm) | | |
|---|---|---|---|
| | Rapid Mixer | Flocculation Basin | Sedimentation Tank |
| 250 | 8.39 | 7.96 | 14.41 |
| 300 | 8.06 | 7.89 | 11.21 |
| 350 | 8.04 | 7.95 | 10.58 |

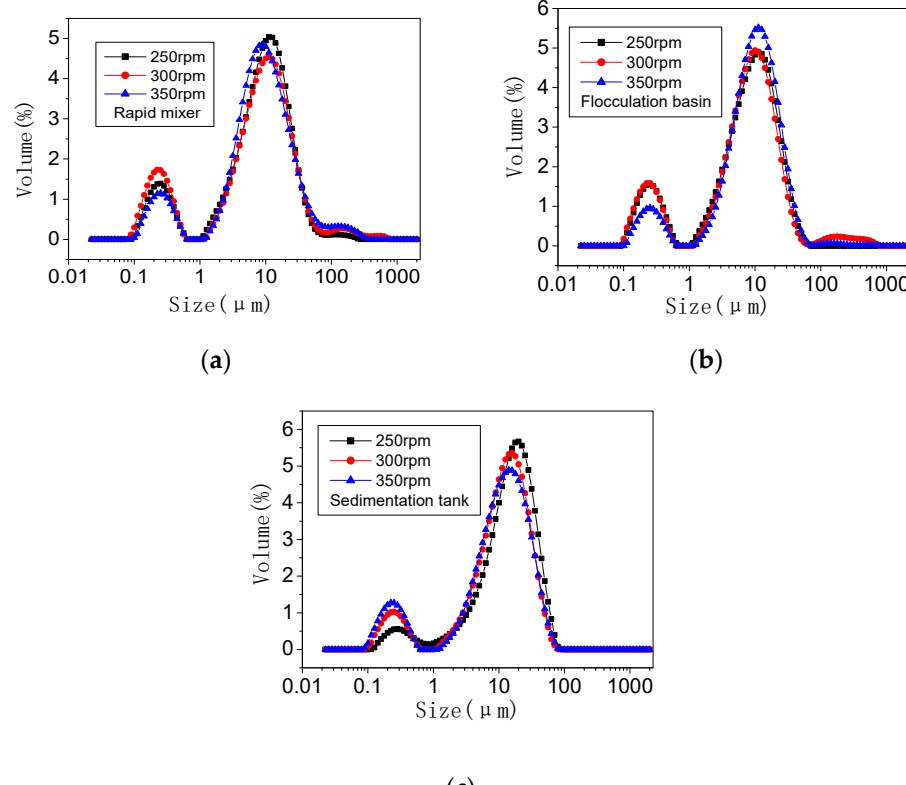

(a)　　　　　　　　　　　　　　　(b)

(c)

**Figure 2.** Floc size distribution with different rapid mixing for three stages. (**a**) rapid mixer; (**b**) flocculation basin; (**c**) sedimentation tank.

### 3.1.2. Removal Efficiency and Zeta Potential

As shown in Figure 3, the K-GN removal efficiency after coagulation remained stable at about 89% in three different rapid mixing conditions. Although Removal efficiency is not significantly influenced by rapid mixing speed, zeta potential increased with increasing rapid mixing speed. Changes in floc characteristics also lead to changes in zeta potential. Rapid stirring speed may change the zeta potential of the colloid, because the rapid mixing could change the floc size and their surface properties. Positive magnesium hydroxide acts as a charge neutralization species. Similar results were also found in the magnesium hydroxide coagulation process for the removal of reactive dyes [18,19]. Zeta potential is important in terms of the impact on steady state floc size and response to increased levels of shear. Under different shear conditions, the break-up flocs may have different physical properties. Floc properties impacted significantly on the overall process efficiency [20]. Electrical charge or colloidal properties of the magnesium hydroxide-reactive orange flocs would be greatly affected. Based on this observation, it can be reasoned that charge-neutralization is one of the mechanisms for destabilization and removal of reactive orange [21]. Normally, slow mixing conditions also affects the flocs break-up and growth. However, in the magnesium hydroxide coagulation process, magnesium hydroxide formed and flocs grew fast, then the larger flocs broke into relatively small particles in the slow mixing period. As previously found [13], floc growth is not significantly influenced by a slow mixing period.

In this research, the effects of slow mixing conditions on floc properties were not considered. As has been mentioned, large flocs breakage and re-growth will happen simultaneously and, consequently, there is less opportunity for the break flocs to re-grow.

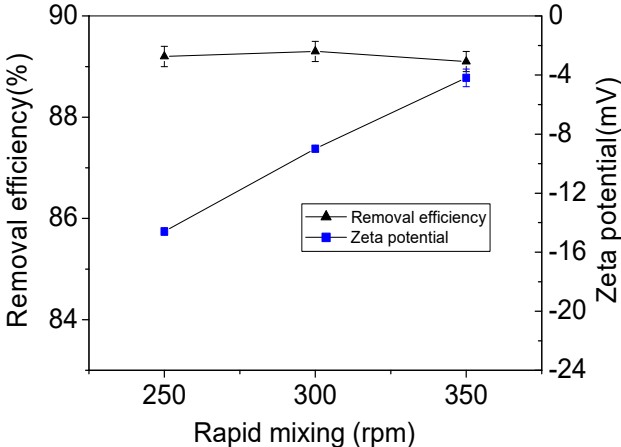

**Figure 3.** Effects of rapid mixing on removal efficiency and zeta potential.

### 3.2. Effect of Flow on Coagulation Performance

For continuous coagulation reaction, the flow determines the retention time of reactants in each reactor. The increase of flow indicates the decrease of retention time. In order to investigate the effect of flow on coagulation performance, continuous experiments were performed under magnesium ion 250 mg/L, rapid mixing speed 300rpm with 30 L/h and 60 L/h. The retention time in the rapid mixing tank was 2min and 1min for flow of 30 L/h and 60 L/h, respectively. As shown in Table 4, the average floc size decreased with increasing influent flow for each operation unit. As for the rapid mixer, the average floc size was 8.06 and 7.25 μm for 30L/h and 60 L/h, respectively. According to earlier results [14], in the stage of rapid mixing, magnesium hydroxide was formed and particles grew in a very short time. A suitable period of rapid mixing was necessary for good coagulation. Reactive orange removal efficiency reached 89% and 83% for 30 L/h and 60 L/h, respectively. The retention time in each unit for 60L/h was shorter than that of 30 L/h, especially in the rapid mixer of 1 min retention time. This is too short to form magnesium hydroxide-reactive orange flocs.

**Table 4.** Average size in different process units.

| Flow (L/h) | Average Size (μm) | | |
|---|---|---|---|
| | Rapid Mixer | Flocculation Basin | Sedimentation Tank |
| 30 | 8.06 | 7.89 | 11.21 |
| 60 | 7.25 | 7.52 | 10.47 |

Figure 4 clearly indicates floc formation and growth in the magnesium hydroxide-reactive orange continuous coagulation system. In the rapid mixer and flocculation basin, average floc size remained stable and flocs aggregated to relatively large flocs in the sedimentation tank. These two figures showed the same trends that the magnesium hydroxide coagulation process was similar to the precipitation process. When magnesium ion is added to the alkaline solution in rapid mixing period, the reaction crystallization process will happen rapidly. Magnesium hydroxide coagulation includes magnesium hydroxide nucleation and a combination of reactive orange into flocs.

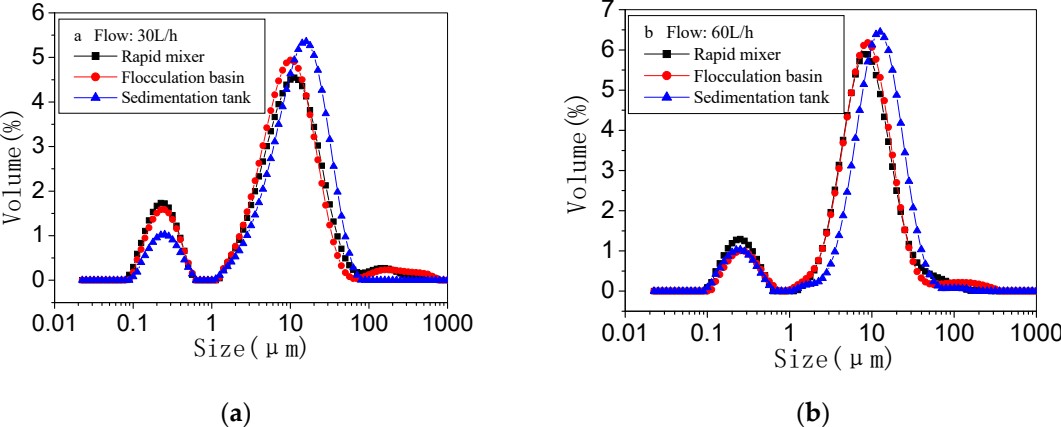

**Figure 4.** Floc size distribution with different flows for three stages. (**a**) 30 L/h; (**b**) 60 L/h.

### 3.3. Images Analysis

The floc size distribution can tell us the percentages of different floc sizes (shown in Figure 4), and the removal efficiency is commonly used to estimate the coagulation performance. In order to gain further insight into the floc and sediment properties, image analysis was used to predict floc characteristics by IX71 digital photomicrography. In Figure 5, the average size of flocs in different reactors is clearly indicated. The floc formation and growth processes were carried out in the rapid mixer and no more aggregation occurred due to the strong repulsion between positively charged particles of magnesium hydroxide in the flocculation basin. In the sedimentation tank, flocs aggregated to form relatively large particles. This is consistent with the findings of FSD value which are shown in Figure 4. As for Figure 5a,d, they clearly show that increasing flow caused a decrease of the average size in the rapid mixer. The average floc size of magnesium hydroxide-reactive orange is 8.06 and 7.25 μm. This is also consistent with the findings of a previous study [14].

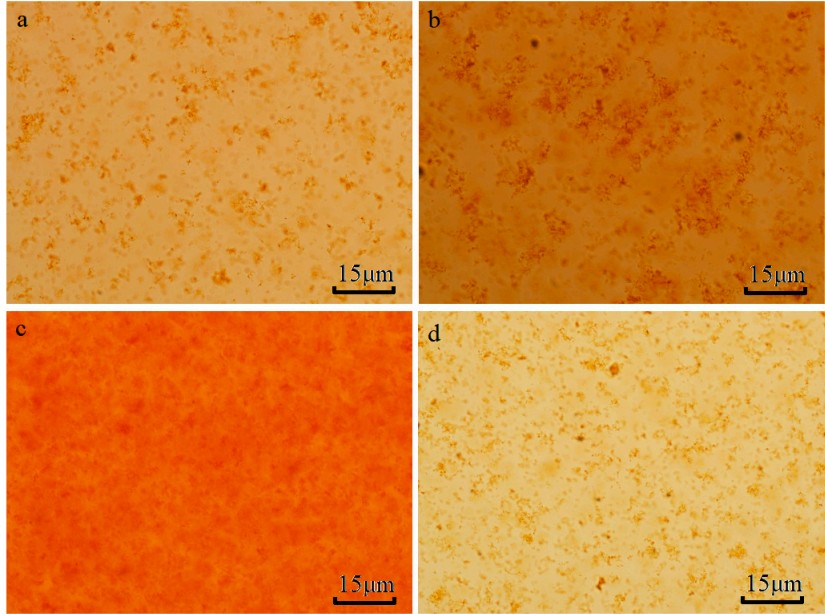

**Figure 5.** Floc image analysis (**a**) rapid mixer for 30 L/h; (**b**) flocculation basin for 30 L/h; (**c**) sedimentation tank for 30 L/h; (**d**) rapid mixer for 60 L/h.

## 4. Conclusions

In this research, the effects of mixing conditions on magnesium hydroxide continuous coagulation performance and floc properties were investigated. Rapid mixing speed plays a significant role in floc

formation and growth. The function of the rapid mixer is to provide a place for the rapid nucleation and precipitation of magnesium hydroxide. In this stage, coagulant and reactive orange formed flocs. In the flocculation basin, small flocs will form more dense flocs, but average floc size remains stable. After the sedimentation process, flocs aggregate to form relatively large flocs. Therefore, appropriate rapid stirring is conducive to the formation of flocculation and the removal of pollutants. Increasing the flow will cause a short retention time; the average floc size decreases from 11.21 to 10.47 μm when flow increases from 30 to 60 L/h. The coagulation behavior indicates that magnesium hydroxide is an effective coagulant for reactive orange removal with an efficiency of 89%.

**Author Contributions:** Investigation, W.L.; data curation, L.W.; writing—original draft preparation, Y.D.; writing—review and editing, J.Z.; funding acquisition, Y.C.

**Funding:** This research was funded by the Technology Research and Development Program of Tianjin, China, grant number 16YFXTSF00390.

**Acknowledgments:** The authors would like to acknowledge the support from Tianjin Key Laboratory of Aquatic Science and Technology.

**Conflicts of Interest:** The authors declare no conflict of interest.

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
