# Peer review of "Effects of Mixing Conditions on Floc Properties in Magnesium Hydroxide Continuous Coagulation Process"

_applsci, doi:10.3390/app9050973_

Round 1

Reviewer 1 Report

Authors addresses an interesting issue a research ambition adequate for a high-end scientific publication.

Three very important factors are not presented in the paper: (1) I could not find any information on the used coagulant dosage (2) Coagulant pH (pH after addition of the coagulant) (3) number fo replicates.

I believe authors have selected one dosage for all experiments, which partially justifies the lack of dosage information. However, if the tests were repeated with a couple of other dosages, it is not obvious that the same observations/conclusions can be made. Especially when discussing the size differences in micrometre scale, replicates are requited to secure repeatability. The need is even more emphases when using the sampling techniques which the authors have used to study the floc size. Fig 3 has some error bars indicating replicates, but information is given.

Fig 3 shows the increase of ZP with the increase of mixing speed. Assuming that the hydrolysis process is full accomplished under all 3 mixing conditions, there is no obvious reason why ZP will be depended on mixing conditions. This observations deserves a more detailed discussion before concluding as the authors done.

It is a generally accepted fact that the slow mixing (flocculation) has the highest influence on flic growth. In fact, the whole concept of having a slow mixing stage is to enhance the floc growth. Authors declare slow mixing has no impact on the floc size based on the observations in table 2. In fact, the floc size is reported to be smaller in flocculation basin than in the rapid mix, which is abnormal. Could it be because the flocs were broken during the transfer from rapid mix basin to the flocculation basin? Or during the sampling?

Author Response

Authors addresses an interesting issue a research ambition adequate for a high-end scientific publication.

Author’s response: 

Authors would like to thank the reviewer for providing careful review and thoughtful recommendation. Responses are provided following each comment in red.

Three very important factors are not presented in the paper: (1) I could not find any information on the used coagulant dosage (2) Coagulant pH (pH after addition of the coagulant) (3) number of replicates.

Author’s response:

The authors would like to thank the reviewer for suggestion. (1) In this paper magnesium ion concentration was 250mg/L and listed in 2.2. Apparatus and procedures. (2) The initial pH was 12 and final pH was 11.4-11.5 after addition of magnesium ion.(3) Each sample was measured  three times and obtained the average results.

I believe authors have selected one dosage for all experiments, which partially justifies the lack of dosage information. However, if the tests were repeated with a couple of other dosages, it is not obvious that the same observations/conclusions can be made. Especially when discussing the size differences in micrometre scale, replicates are requited to secure repeatability. The need is even more emphases when using the sampling techniques which the authors have used to study the floc size. Fig 3 has some error bars indicating replicates, but information is given.

The authors agree with the reviewer. In fact, three dosages of magnesium ion (225, 250, 300mg/L) were used for the experiments. The removal efficiency increased with magnesium ion increase. Because this paper was mainly discussed the mixing condition effect, dosages of magnesium ion will be discussed in another paper. We performed at least three hours and all samples of flocs were taken from rapid mixer, the third flocculation basin and sedimentation tank every ten minutes and floc size distribution were measured by Mastersizer 2000 (Malvern,UK) and each sample was measured three times and obtained the average results. When floc size distribution has the same trends at each of units,the data was chosen from one of them to formed Figures. 

Fig 3 shows the increase of ZP with the increase of mixing speed. Assuming that the hydrolysis process is full accomplished under all 3 mixing conditions, there is no obvious reason why ZP will be depended on mixing conditions. This observations deserves a more detailed discussion before concluding as the authors done.

Rapid stirring speed may change Zeta potential of colloid, because the rapid mixing could change the floc size and their surface properties (especially surface charge). When magnesium ion was added to the sample water, an increase in pH upon alkalization will result in the precipitation of magnesium hydroxide. The primary nucleation process of magnesium hydroxide will affect the floc formation time of colloids in the simulated water samples.

It is a generally accepted fact that the slow mixing (flocculation) has the highest influence on flic growth. In fact, the whole concept of having a slow mixing stage is to enhance the floc growth. Authors declare slow mixing has no impact on the floc size based on the observations in table 2. In fact, the floc size is reported to be smaller in flocculation basin than in the rapid mix, which is abnormal. Could it be because the flocs were broken during the transfer from rapid mix basin to the flocculation basin? Or during the sampling?

Magnesium hydroxide coagulation process is different from conventional coagulants such as aluminum and iron salts. Rapid mixing brings the reactants together and homogenizes the solution. In rapid stirring process, it will cause nucleation of magnesium hydroxide. Magnesium hydroxide coagulation process had two stages including fast floc formation and growth of flocs and then the larger flocs break into relative small particles. Magnesium hydroxide coagulation flocs show breakage in slow mixing periods. As can be seen also in Table 2, average particle size was 8.06 and 7.89 μm in rapid mixer and flocculation basin when rapid mixing speed was 300rpm and slow mixing speed was 80rpm. In generally, the flocs will grow in the flocculation basin, but it seems that floc size decreases in flocculation process. In fact, during flocculation process, particles larger than 11.25 μm accounted for 4.54% and 4.9% in rapid mixer and flocculation basin, respectively. Particles smaller than 1 μm also decreased in slow mixing period. Flocs growth and breakage will happen at the same time. It showed poor ability to aggregate together. Magnesium hydroxide precipitation has a positive superficial charge that repulsive forces tend to stabilize the suspension and prevent particle agglomeration.

Reviewer 2 Report

Please, see the attached document. 

Author Response

The article deals with the study of the effects of mixing conditions on the floc properties during the coagulation of a dye polluted wastewater. The study also brings new information to the literature, since it is almost a full-scale application; the study was carried out, differently from what was already published, in a continuous manner (novelty of the study). Below, my comments are reported:  

1.  Introduction. This section needs to be strengthened with the addition of additional articles on coagulation/flocculation processes applied to waste water. For the scope, I recommend these articles:  

De Feo G., Galasso M., Landi R., Donnarumma A., De Gisi S. (2013). A comparison of the efficacy of organic and mixed-organic polymers with polyaluminium chloride in chemically assisted primary sedimentation (CAPS), Environmental Technology, Volume 34(10), pages 1297-1305

(https://tandfonline.com/doi/full/10.1080/09593330.2012.745622);

De Feo G., De Gisi  S., Galasso M. (2008). Definition of a practical multi-criteria procedure for selecting the best coagulant in a chemically assisted primary sedimentation process for the treatment of urban wastewater. Desalination, Volume  230(1-3),  pages  229-238

(https://www.sciencedirect.com/science/article/pii/S0011916408003184);  

Rizzo L., Lofrano G., Grassi M., Belgiorno V. (2008). Pre-treatment of olive mill wastewater by chitosan coagulation and advanced oxidation processes. Separation and Purification Technology 63(3), 648-653 (https://www.sciencedirect.com/science/article/pii/S1383586608002761).  

Author’s response:

The authors would like to thank the reviewer for suggestion. These articles are useful for this research, the authors choose one of them as the reference. As for “Pre-treatment of olive mill wastewater by chitosan coagulation and advanced oxidation processes”, this paper used chitosan as coagulant for pre-treatment of olive mill wastewater. For “Definition of a practical multi-criteria procedure for selecting the best coagulant in a chemically assisted primary sedimentation process for the treatment of urban wastewater”, jar tests were carried out to determine coagulant conditions for urban wastewater treatment.

2. Materials and methods. The test modalities should be better explained. For this purpose, a new table should be added to show how the contact time/rotation speed varies with the flow rate and with the dye mass flow rate. Furthermore, this table must also contain a row showing the contact time and rotation speed values according to the Standards for this type of test.  

Some further considerations: why was such a high rotation speed (80 rpm) adopted in the flocculation phase? Generally, lower values are reported (e.g., 30 rpm).  

 The authors agree with the reviewer. A new table was added in this paper.

Table 2. Operational conditions of coagulation process

Flux

Rapid mixer

Flocculation basin

Sedimentation tank

30L/h

speed

time

speed

time

time

250rpm

2min

80rpm

24min

60min

300rpm

350rpm

60 L/h

300rpm

1min

80rpm

12min

30min

Magnesium hydroxide precipitation has a positive superficial charge that repulsive forces tend to stabilize the suspension and prevent particle agglomeration. In our previous study, slow mixing speed of 40, 60, 80 and 100rpm was chosen and they also showed that floc will break when. This is consistent with the findings of our former studies (Zhao, J., Shi, H., Liu, M., Lu, J. & Li, W. 2017 Coagulation-adsorption of reactive orange from aqueous solution by freshly formed magnesium hydroxide: mixing time and mechanistic study. Water Science and Technology, 71 (9), 1310-1316, and Zhao, J., Su, R., Guo, X., Li, W., Feng, N. 2014 Role of mixing conditions on coagulation performance and flocs breakage formed by magnesium hydroxide, J. Taiwan. Inst. Chem. E. 45 : 1685–1690.

3. Results and discussion. Table 2 shows how the size of the flock collected from the flocculation tank is smaller than that of the flock in the rapid mixing tank. This result is not justified, since generally, the more calm conditions in the flocculation basin favour the growth of  the floc, whichthen finds completion  in  the sedimentation  tank. Therefore, the question is: why does the floc not grow in the flocculation basin? Is it due to the excessive rotation speed (80 rpm) that was adopted?

As can be seen also in Table 2, average particle size was 8.06 and 7.89 μm in rapid mixer and flocculation basin when rapid mixing speed was 300rpm and slow mixing speed was 80rpm. In generally, the flocs will grow in the flocculation basin, but it seems that floc size decreases in flocculation process. In fact, during flocculation process, particles larger than 11.25 μm accounted for 4.54% and 4.9% in rapid mixer and flocculation basin, respectively. Particles smaller than 1 μm also decreased in slow mixing period. Flocs growth and breakage will happen at the same time. It showed poor ability to aggregate together. 

Reviewer 3 Report

Physicochmical condition and hydrodynamic conditions are main factor to control the properties of resulted flocs. This is well-known and along this line this study is reported. Methods are already established in the previous study. Content si reliable.

Minor suggestions are cited reference are not appropreate. The author can found another source of references in the reference list of,

Determination of rate of salt-induced rapid coagulation of polystyrene latex particles in turbulent flow using small stirred vessel” (Manuscript ID: colloids-396128). 

especially,

 Kobayashi, M.; Adachi, Y.; Ooi, S. Breakup of fractal flocs in a turbulent flow. Langmuir 1999, 15, 4351–4356.

describe two asspects.

For the formation process., the way of analysis is described already 30 years ago in:

         Adachi,Y.;CohenStuart,M.A.;Fokkink,R. Kinetics of turbulent coagulation studied by means           of end-over-end rotation. J. Colloid Interface Sci. 1994, 165, 310–317.          

 and  for the breakup, the properties which will be related to rheology are discussed in

         Kobayashi, M.; Adachi, Y.; Ooi, S. Breakup of fractal flocs in a  turbulent flow. Langmuir                 1999, 15, 4351–4356.
But after that that much more developments have been done. It is obvious that author neglect all of them.    
         For instance,

         1. G. Frappier, B.S. Lartiges, S. Skali-Lami, Floc cohesive force in reversible aggregation: a           Couette laminar flow investigation. Langmuir, 2010, 26 (13), 10475-10488.

demonstrate similar size disributiion of flocs. 

the pattern of breakup is described clearly in
           1. K. Higashitani, N.Inada, and T.Ochi, Colloids Surf. A Physicochem. Eng. Asp.,1991, 56, 13-91

If you think restructuring, probably author can find the hint of consideration in

1. Y. Adachi and S.  Ooi, Geometrical structure of a floc. Colloid Interface Sci., 1990, 135, 374-384.

Author Response

Author’s response:

The authors would like to thank the reviewer for suggestion. These literature are very good for coagulation process and floc formation and breakup. Such as: (G. Frappier, B.S. Lartiges, S. Skali-Lami, Floc cohesive force in reversible aggregation: a Couette laminar flow investigation. Langmuir, 2010, 26 (13), 10475-10488.) This parer used a simple theoretical model to describe the limiting size of aggregates attained at steady state under given shear conditions. Magnesium hydroxide coagulation process is similar to the precipitation process which includes magnesium hydroxide nucleation and combination of reactive orange into flocs. In the early stage of rapid mixing, magnesium hydroxide was formed and particles grew in very short time (90s). Magnesium hydroxide coagulation process is different from conventional coagulants such as aluminum and iron salts. Magnesium hydroxide coagulation process had two stages including fast floc formation and growth of flocs and then the larger flocs break into relative small particles. In this paper, the authors discussed the role of mixing conditions on floc properties of magnesium hydroxide coagulation process. The theoretical analysis will be discussed in next paper. Thank you very much again for your suggestions.

Round 2

Reviewer 2 Report

First of all, the Authors did not prepare the file "reply to reviewers" by which they responded to the observations in a timely manner. I had difficulty in revising the manuscript. 

Then, some of my questions remained unanswered. They are list below: 

1) With reference to the test modalities (materials and methods), why was such a high rotation speed (80 rpm) adopted in the flocculation phase, where generally lower values are considered (30 rpm)? For the scope, I suggest to consider the following reference that contains the values generally adopted for the phase of coagulation (rapid mixing), flocculation and sedimentation: Standard methods for the Examination of Water and Wastewater, 19th edition, American Public Health Association/American Water Works Association/ Water Environment Federation, Washington DC, 1995.

2) The Authors did not answer my question: why does the floc not grow in the flocculation basin? Is it due to the excessive rotation speed that was adopted (and equal to 80 rpm)?

Please, without being superficial, give a calm answer to my questions. 

Author Response

Author’s reply to reviewers’ comments

First of all, the Authors did not prepare the file "reply to reviewers" by which they responded to the observations in a timely manner. I had difficulty in revising the manuscript. 

Then, some of my questions remained unanswered. They are list below: 

1) With reference to the test modalities (materials and methods), why was such a high rotation speed (80 rpm) adopted in the flocculation phase, where generally lower values are considered (30 rpm)? For the scope, I suggest to consider the following reference that contains the values generally adopted for the phase of coagulation (rapid mixing), flocculation and sedimentation: Standard methods for the Examination of Water and Wastewater, 19th edition, American Public Health Association/American Water Works Association/ Water Environment Federation, Washington DC, 1995.

Author’s response:

The authors would like to thank the reviewer for suggestion. There are two reasons for choosing 80rpm as slow mixing speed. (1) For our continuous experiments, each of flocculation basin is 4L, mixing speed lower than 80rpm is not stable operation. (2) Based on jar tests experiment, floc growth is not significantly influenced by slow mixing period (30-80rpm) for magnesium hydroxide coagulation process.

2) The Authors did not answer my question: why does the floc not grow in the flocculation basin? Is it due to the excessive rotation speed that was adopted (and equal to 80 rpm)?

Please, without being superficial, give a calm answer to my questions. 

Author’s response:

Normally, flocs will grow in flocculation basin (slow mixing condition). But in magnesium hydroxide coagulation process, floc size tended to stable or slightly broken. The authors agree with the reviewer. 80rpm (excessive rotation speed) maybe high and cause floc break. In our previously found, mixing speed (30-80rpm) will cause floc break also. In next continuous experiments periods, we will adjust operation conditions again including rapid/slowing mixing speeds, reactor volumes, retention times. Thank you very much for your comments again. 

Round 3

Reviewer 2 Report

The Authors have answered my questions.